# Hardware Scaling Trends and Diminishing Returns in Large-Scale Distributed Training

Dramatic increases in the capabilities of neural network models in recent years are driven by scaling model size, training data, and corresponding computational resources. To develop the exceedingly large networks required in modern applications, such as large language models (LLMs), model training is distributed across tens of thousands of hardware accelerators (e.g. GPUs), requiring orchestration of computation and communication across large computing clusters. In this work, we demonstrate that careful consideration of hardware configuration and parallelization strategy is critical for effective (i.e. compute- and cost-efficient) scaling of model size, training data, and total computation. We conduct an extensive empirical study of the performance of large-scale LLM training workloads across model size, hardware configurations, and distributed parallelization strategies. We demonstrate that: (1) beyond certain scales, overhead incurred from certain distributed communication strategies leads parallelization strategies previously thought to be sub-optimal in fact become preferable; and (2) scaling the total number of accelerators for large model training quickly yields diminishing returns even when hardware and parallelization strategies are properly optimized, implying poor marginal performance per additional unit of power or GPU-hour.

## 1 Introduction

Empirical compute-optimal scaling laws show that the performance of large neural networks increases jointly with: the model size, volume of training data, and the amount of allocated training compute (i.e. FLOPs) (Hoffmann et al., 2022a;b; Kaplan et al., 2020; Tay et al., 2023; Porian et al., 2024). These scaling trends have naturally incentivized rapid increases in model size over the past decade in pursuit of state-of-the-art performance across a variety of applications in natural language processing and computer vision (Chowdhery et al., 2023; Zhang et al., 2022; Dehghani et al., 2023).

The increased size of state-of-the-art neural networks, containing hundreds of billions of parameters, yields larger computational workloads and memory requirements during training. In this regime, the memory requirements from increasing numbers of model parameters and large-batch sizes are often sufficiently large such that a single model cannot fit inside the memory of a single GPU accelerator. To leverage the increased processing power and memory of additional devices, the largest workloads necessitate distribution across hundreds to thousands of hardware accelerators (i.e. GPUs, TPUs). Training in these settings requires complex parallelization strategies for distributing data, model parameters, activations, gradients and optimizer states across accelerators –

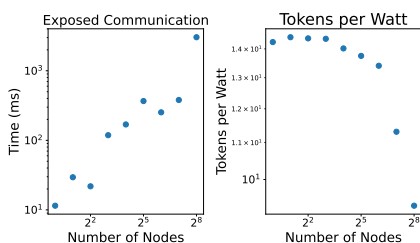

Figure 1: As scale of distributed training increases, the power efficiency decreases while the amount of exposed communication increases.

discussed in more detail in §2 (Rasley et al., 2020; Shoeybi et al., 2019; Zhao et al., 2023; Li et al., 2020; Ryabinin et al., 2023).

As the number of devices required for large-scale neural network training has increased, the underlying cost dynamics of communication and computation have changed. Previously, the high arithmetic intensity of deep learning models meant that most workloads were compute-bound – i.e. runtime was dominated by component convolutions or matrix multiplications (Jouppi et al., 2017; Micikevicius et al., 2017). However, in large-scale distributed training parallelization of compute and synchronization across massive pools of accelerators increases the communication volume between accelerators and nodes required to synchronize parameters, gradients, and optimizer states. Limitations on the efficiency of network fabrics and some collective communication algorithms in-

cur a high marginal cost to scale and bound the degree of model *sharding*, where components of a single model are distributed across devices. In contrast to traditional high-performance computing workloads executed on CPU-based architectures, where lower arithmetic intensity and relatively slow-speed of computation overlapping communication and computation more possible (Lee et al., 2010; Hill & Marty, 2008; Asanovic et al., 2006; Amdahl, 1967), deep learning training becomes unavoidably *communication-bound* at scale. In this work, we show how this limits the extent to which model size and parallelization across additional accelerators can be increased while still producing improvements in overall throughput – due to the additional communication.

In experiments analyzing this trend across hardware platforms, we further show that improvements in accelerator computation performance have outpaced improvements in memory bandwidth and network performance, suggesting that communication-boundedness worsens as a function of recent improvements in hardware efficiency. Together, these factors demonstrate that large scale deep learning computation suffers from significantly *diminishing returns when horizontally scaled across larger number of devices* in massively distributed settings.

While there are stable distributed training recipes that perform well at large scale, their scaling properties are not yet well characterized. In this work, we contribute the following:

- A **large-scale empirical study** of distributed training across hardware setups, model sizes, and parallelism strategies, characterizing the scaling properties of sharded training

- Demonstration of **diminishing returns for scaling the number of accelerators** for training as measured by words-per-second throughput, due to increasing communication overhead

- **Analysis of real-world cost metrics** which shows that the total GPU power draw and available FLOPS scale linearly with the number of devices, despite diminishing returns in throughput; resulting in reduced power efficiency (e.g. tokens per watt) and lower hardware utilization (See Figure 1)

- Evidence that **model parallelism yields improved global throughput** despite prior work (Hagemann et al., 2023; Narayanan et al., 2019) and conventional knowledge suggesting that model parallelism lowers hardware utilization

- A **study across hardware generations** demonstrating that future improvements in computational throughput will only marginally improve overall performance absent network fabric advancements and increased accelerator memory capacity.

## 2 PRELIMINARIES

### 2.1 ACCELERATORS AT SCALE

Large neural networks are trained in computing clusters consisting of thousands of interconnected GPUs characterized by compute power, on-board memory, and interconnects. Several technologies are used to interconnect GPUs (Recio et al., 2007; Shainer et al., 2011) — each have tradeoffs between network size, bandwidth, latency, and cost. This results in a hierarchical partitioning of the network, with communication within groups achieving higher bandwidths or lower latencies than communication across groups. In Figure 2, we provide an illustration of a sample architecture for a GPU-node based datacenter cluster.

For example, NVIDIA GPUs that are grouped together within a node (typically with 8 accelerators[1]) might be connected via technologies such as NVLink or NVSwitch (Nvidia, 2024), which can provide an order of magnitude higher bandwidth than the network fabric used to communicate across nodes. Within the network itself, the topology of the switches can further favor the communication within certain groups of nodes, such as within racks or "rails."

These accelerator characteristics – compute, memory and network – have evolved at different rates over time based on technical barriers and demand for usage. In turn, neural architectures and software systems have been designed to adapt to the limits of the hardware. With the increase in cluster size driven by larger models and data, communication has often been such a bottleneck, and many new training algorithms have thus emerged (Shazeer et al., 2017; Fedus et al., 2022).

---

[1]NVIDIA's DGX-1 P100 set early precedent.

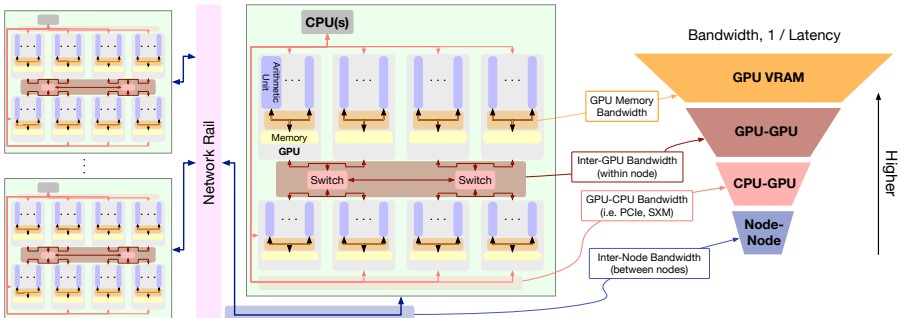

Figure 2: An abbreviated representation of the hierarchy of connections between components of a datacenter GPU system with multiple 8-GPU nodes connected via a fabric. Connections between components include: (1) *GPU VRAM*: memory (VRAM) buses to compute units, (2) *CPU-GPU*: PCIe or SXM buses from CPU to GPU, (3) *GPU-GPU*: GPU-to-GPU connections within a node, and (4) *Node-Node*: connections between nodes. Also represented is the relative performance of each of these connections.

## 2.2 ALGORITHMS FOR DISTRIBUTED TRAINING

Most distributed training schemes optimize for transparency to practitioners, thus preserving the abstraction of single-device training, where one instance of a model processes a minibatch of data such that every input sample interacts with every parameter of the model. A key decision in distributed training is how to map model and data components onto GPU hardware: via replication or sharding. Communication between GPUs is required to aggregate values that must interact with each other or to preserve synchronization between replicas of the same underlying values.

Below, we provide a brief taxonomy of distributed training algorithms. These algorithms are not mutually exclusive – distribution across multiple "dimensions" can be combined.

**Data parallelism** (Dean et al., 2012) consists of replicating model parameters (and optimizer states) across GPUs, but shards input minibatches across devices. After performing local forward and backward passes on their allocated minibatches, GPUs exchange and accumulate their partial gradients, thus obtaining an identical global gradient and ensuring an identical model update. This communication pattern is named after the MPI (Walker & Dongarra, 1996) collective `AllReduce`.

When models grow too large to fit on a single device, methods such as Fully-Sharded Data Parallelism (FSDP) (Xu et al., 2020) and DeepSpeed-Zero (Rasley et al., 2020) shard optimizer states, gradients, and model parameters across data parallel groups. However, since every input sample must interact with every parameter, they are required to temporarily re-materialize each parameter on-the-fly on all devices during the forward and backward passes. In contrast to communication of gradients, which can be performed concurrently with the backward pass before a model update, "gathering" operations, which are required to gather parameters for the forward pass, delay execution of computation. An inverse operation must be performed to partial gradients during the backward pass to update parameters. These two MPI primitives are known `AllGather` and `ReduceScatter`.

**Model parallelism** shards model parameters across GPUs; each shard operates on the same minibatch simultaneously. In this setting, activations and their respective gradients are sent across GPUs.

- **Tensor Parallelism** (Shoeybi et al., 2019; Shazeer et al., 2018) Some layers (e.g. linear) within models can be sharded along their hidden dimensions, leveraging linear algebra properties to slice weight matrices in a way that maximizes data locality and allows for mostly independent computation before a final `AllReduce` step to re-synchronize activations. As the full set of activations are required for computation with the subsequent layer, Tensor Parallelism introduces *blocking communication* for synchronization of interemediate activations across model parallel groups. *Sequence parallelism*, for instance, is an analog of FSDP's changes to data parallelism: instead of replicating activations, they remain sharded and only get gathered and scattered as needed Li et al. (2023).

- **Pipeline Parallelism** (Huang et al., 2018; Harlap et al., 2018) Models can also be sharded along their layerwise depth, with layers being partitioned into "stages" and stored on different devices; activations are then forwarded from one device to the next while they traverse a model.

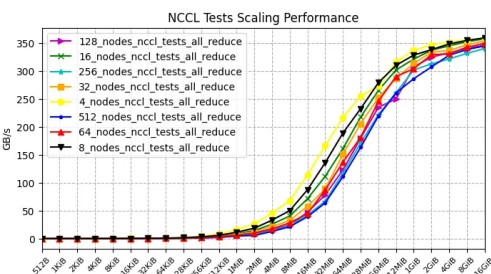 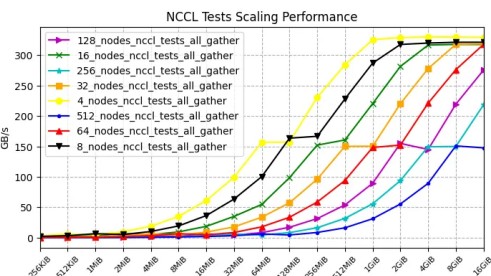

(a) Bandwidth of `AllReduce`, which uses a tree algorithm and scales well with number of nodes (i.e. higher bandwidth).

(b) Bandwidth of `AllGather`, which uses ring algorithms; scales poorly with the number of nodes (i.e. lower bandwidth).

Figure 3: Bandwidth measurements in GB per second of NCCL primitives on DGX H100 servers with eight GPUs per node, connected with InfiniBand, across world sizes from 4 to 512 nodes.

For all devices to be active at once, an input minibatch is split into microbatches which are then staggered and pipelined according to a schedule. A "bubble" (Hennessy & Patterson, 2017), in which devices remain idle, reduces the efficiency of pipelining.

**Communication-Computation Overlap and Communication Exposure** Moving data over networks between accelerators utilizes distinct GPU resources unrelated to computation (e.g., dedicated copy engines, NVLink/NVSwitch) and can execute in parallel with computation. Overlapping communication and computation maximizes distributed training efficiency – it facilitates hiding communication latency, leading to near-perfect scaling. We define communication as *exposed* when it is not hidden by simultaneous computation, leaving a GPU's compute resources under-utilized.

### 2.3 COMMUNICATION PRIMITIVES AND LIBRARIES

Modern deep learning frameworks (Paszke et al., 2019; Abadi et al., 2015; Bradbury et al., 2018) leverage specialized collective communications libraries, such as NCCL[2] , RCCL[3] or XLA[4]. These libraries may contain multiple algorithms for each collective communication primitive. `AllReduce` being a reduction, has both a "ring" and a "tree" algorithm, the former being bandwidth-balanced but suffering from latency increasing linearly with the number of devices, and the latter being suboptimal in bandwidth utilization but logarithmic in latency. `AllGather` and `ReduceScatter`, which are both used in parameter rematerialization for FSDP and DeepSpeed-Zero, can only use ring algorithms as all buffers must be delivered to all devices – and quickly become latency-bound as the number of devices increases, as shown in Figure 3.

### 3 EXPERIMENTAL METHODOLOGY

In the following sections, we investigate the effects of scaling training workloads on computation and communication volume; and the impact of scale on end-to-end system performance. In particular, we conduct experiments across: distributed parallelization strategies, numbers of accelerators (i.e. GPU device world size), hardware generation, model sizes, and input shapes (i.e. context length). Additional details on hardware and framework configurations are provided in Appendix C.

**Model Architectures** We conduct our investigation focusing on the Llama-2 architecture of decoder-only transformers (Dubey et al., 2024; Touvron et al., 2023), as a representative architecture for state-of-the-art neural large language models. We utilize the AdamW optimizer (Loshchilov & Hutter, 2019; Kingma & Ba, 2015) and train on examples with a context length of 4096 and tokenized with a vocabulary of 32K; with data sampled from Wikipedia and StackExchange.

---

[2]https://github.com/NVIDIA/nccl

[3]https://github.com/ROCm/rccl

[4]https://github.com/openxla/xla

We primarily investigate training models at the 7B parameter scale, and conduct additional experiments on the effects of architecture scaling at 1B, 7B, 13B, and 70B parameters.

**Hardware Configuration** We evaluate distributed training on datacenter clusters containing 8-GPU NVIDIA DGX nodes from the Ampere (80GB A100) and Hopper (80GB H100) architectures. Intra-node GPU communication is occcurs via fully connected second and third generation NVLink with NVSwitch, respectively. Inter-node communication occurs over an Infiniband fabric with 200 GB/s and 400 GB/s per-node bandwidth, respectively. We conduct our primary experiments on hardware scales between 1 and 32 eight-GPU nodes, with additional experiments up to 256 nodes, or 2,048 GPUs – to simulate scales for modern pretraining.

**Parallelization Strategies** We explore the space of parallelization strategies used to distribute the training workload across GPU nodes. We examine data, tensor, and pipeline parallelization strategies (colloquially known as 3D parallelism as described by Shoeybi et al. (2019); Rasley et al. (2020) and used in Dubey et al. (2024); BigScience Workshop (2022)). To address the necessary memory overhead of training large models, models are trained with Fully-Sharded Data Parallelism without additional parameter resharding during the forward pass (i.e. FSDP, Zhao et al. (2023)) as is used in Llama-3.1 training.

We examine a range of group sizes for tensor and pipeline parallel strategies for model parallelism as described in Section 2, ranging from group sizes of 1 (i.e. single GPU training with no parallelization) up to group sizes of 16 (i.e. requiring parallelism groups across multiple nodes).

**Performance Metrics** To understand both the effects of hardware and model scaling on end-to-end global and local per-device performance hardware utilization, we examine the following variety of performance and efficiency indicators:

- *Throughput* is the rate at which examples are processed. We compute the estimated per-device *words per second* (WPS) and the global words per second across all devices.
- *Computational and communication load* can be measured as the total execution time for CUDA and NCCL kernels, respectively. We calculate the total computation and communication load by aggregating and flattening CUDA and NCCL kernels from PyTorch execution traces.
- *Communication efficiency* can be measured as the extent to which communication kernels are exposed or overlapped with concurrent computation.
- *Hardware utilization* can be measured as the number of floating point operations per second (FLOPS); alternatively, as Model FLOPS Utilization (MFU, Chowdhery et al. (2023)) which is the observed FLOPS as a percentage of the hardware's theoretical maximum.
- *Power utilization* can be measured as the per-GPU power draw, and estimated as the power utilization across all devices. We measure the average power draw with NVML[5].

We compute these metrics over 60 training iterations; discarding the first 10 iterations to allow for GPU memory allocations and stabilization of performance during the initial training iterations, and aggregate metrics for the last 50 iterations.

# 4 EFFECTS OF SCALING: PARALLELIZATION, HARDWARE, & MODEL SIZE

In this section, we examine the effects of scaling neural network architecture sizes, their underlying hardware platform (i.e. number of GPU devices), and the parallelization strategies used to distribute model training onto said hardware platforms.

## 4.1 SCALING DATA PARALLELISM

In Figure 4, we examine the effects of scaling data parallel training across increasing numbers of accelerators from 8 GPUs up to 2048 GPUs. In this setting, each device carries a data parallel replica and trains Llama-7B model with a constant local batch size of 2. As expected, increasing the number of devices yields increases in overall global throughput as the global batch size increases.

---

[5]https://developer.nvidia.com/management-library-nvml

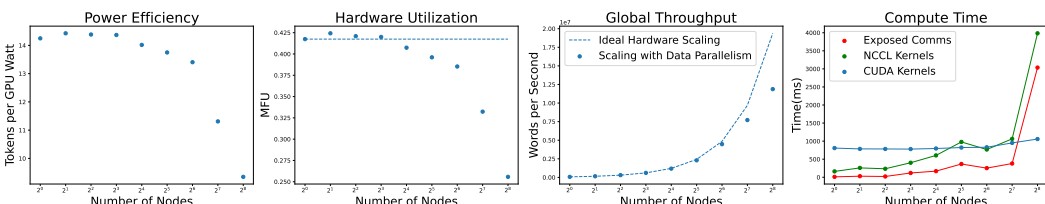

Figure 4: In FSDP training of Llama-7B, scaling the number of nodes and data parallel replicas *reduces hardware utilization and power efficiency*; due to increasing exposed communication derived from increases in the size of communication kernels relative to fixed size computation kernels. Global throughput observes sublinear scaling despite approximately linear increases in the total power utilization with number of nodes. "Ideal Hardware Scaling" corresponds to expected throughput should introduction of additional accelerators produce linear increases in throughput.

At small scales, when training using a limited number of devices, the cost of collective communication kernels is low relative to the cost of computation – and non-blocking communication from FSDP can be hidden by executing data transfer and computation operations concurrently.

However, as discussed in Section 2 and in Figure 3, increasing degree of data parallelism also incurs the cost of larger collective communication operations needed to materialize parameters via `AllGather` during the forward pass and to update gradients during the backward pass via `ReduceScatter`. As observed in Figure 4, we observe that the execution time for NCCL communication kernels and volume of exposed communication scales with the number of compute nodes; matching the expected behavior observed for the communication collectives seen in Figure 3b.

While the communication volume scales with node count, the per-device CUDA computation kernels execution time remains constant and is dominated by communication. As a result, the exposed communication is unavoidable at scale and the hardware utilization decreases as there is insufficient computation to saturate the GPUs while waiting for the execution of larger communication kernels – this results in reductions the marginal speedup of global throughput and decreased local throughput as the number of devices increases.

These observations are contrary to conventional wisdom which often assumes `AllGather` and `ReduceScatter` operations are non-blocking operations that can be overlaid with computation; and data parallelism can be introduced with limited additional costs. Instead, we observe that a majority of communication becomes exposed at large-scales resulting in long periods of GPUs remaining idle.

While the per-device throughput scales sublinearly with the number of devices, the total power utilization scales approximately linearly resulting in substantially worse real-world efficiency in GPU-hours and energy impact (i.e. fewer tokens processed per watt). When scaling from 128 to 2048 GPUs, the observed TFLOPS and words-per-second throughput decrease by 37.22% due to reduced hardware utilization from exposed communication. Despite operating at lower arithmetic intensity, the per-GPU power draw is roughly constant regardless of the arithmetic intensity – only decreasing by 5.87% from 658W to 620W. As a result, the overall power efficiency of the system likewise decreases with hardware scale as seen in Figure 4.

### 4.2 Scaling Model Parallelism

Model parallelism is commonly used as a technique to reduce the memory pressure of very large models which cannot fit in a single GPU device by sharding individual layers across multiple devices Dubey et al. (2024); Zhang et al. (2022); Team et al. (2023).

Furthermore, model parallelism provides the additional benefit of reducing the sizes of the data parallel groups; as separate data parallel replicas are maintained for each model parallel group (i.e. data parallel collectives are executed over world sizes of $\frac{\text{Number of Devices}}{\text{Total Degree of Model Parallelism}}$, rather than over the Total Number of Devices) – where Total Degree of Model Parallelism is the product of Tensor and Pipeline parallelism group sizes.

As such, we observe in Figure 5 that small degrees of total model parallelism (i.e. model or pipeline parallel degrees of 2 or 4) yields reductions in the amount of *exposed communication*, as the `AllGather` and `ReduceScatter` operations are applied over a smaller data parallel groups and

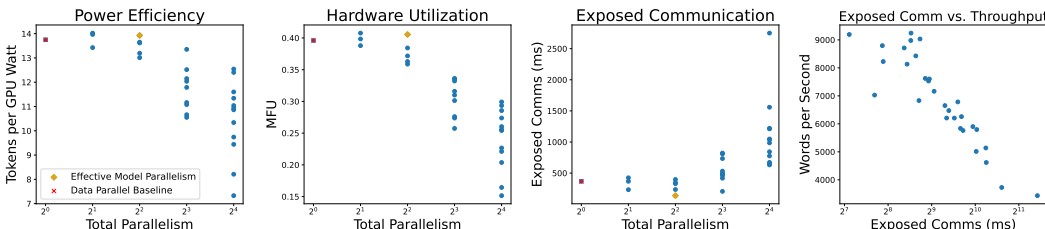

Figure 5: In model parallel training of Llama-7B with a fixed global batch size (512) and fixed number of accelerators (256 GPUs), there exist model parallel strategies that *increase* training throughput, hardware utilization, and power efficiency by reducing the total exposed communication; which is strongly negatively correlated with throughput.

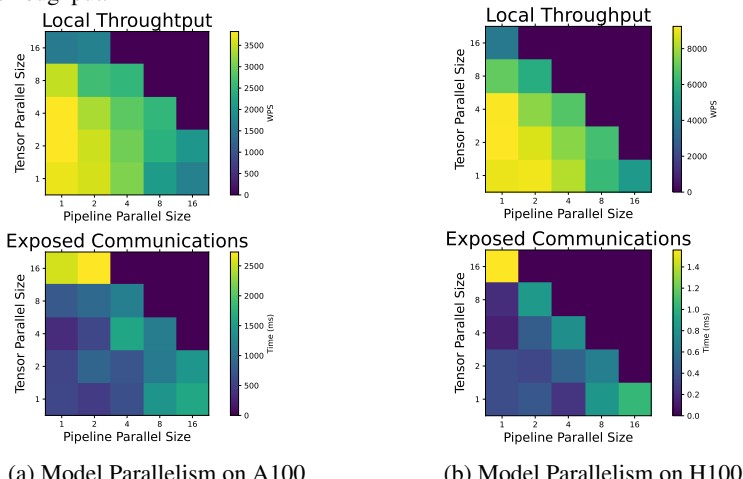

(a) Model Parallelism on A100      (b) Model Parallelism on H100

Figure 6: *Model Parallelism Improves Throughput.* Increasing degree of either tensor and pipeline model parallelism yields improved throughput and less exposed communications compared to the data parallel baseline (i.e. Tensor and Pipeline Parallel Size of 1).

the `AllReduce` operations introduced by Tensor Parallelism exhibit better scaling properties. This is contrary to previous work which often suggests that model parallelism approaches yield lower hardware utilization relative to data parallel baselines (Hagemann et al., 2023; Narayanan et al., 2019) due to the increased total number of communication operations and introduces blocking communication operations to synchronize partial sums of activations required for model parallelization.

We find that there exist effective non-trivial model parallel strategies that: reduce exposed communication, increase hardware utilization and power efficiency. In Figure 6, we find that both tensor and pipeline parallelism exhibit this behavior, in which model parallelism reduces the exposed communication volume and increases word-per-second throughput performance improves relative to the data parallel baseline when utilizing model parallelisms to reduce communication overhead. In Appendix D, we find that as hardware utilization decreases due to low arithmetic intensity or large collective communications, the amount of viable model parallelism strategies increases.

Notably, there is a limit to the extent to which model parallelism reduces exposed communication and improves throughput – as the `AllReduce` kernels required for Tensor Parallelism and bubbles introduced by pipeline parallelism grow with the degree of model parallelism. These communication costs become especially large when the parallelism occurs over multiple nodes as it much rely on slower internode fabric (e.g. InfiniBand, see Figure 2) for communication – as noted in Figure 6, where there is substantial increases in exposed communication for tensor and pipeline parallelism strategies which are sharded at larger than 8 devices (i.e. across multiple nodes).

### 4.3 SCALING THE HARDWARE WORLD SIZE

In Section 4.1, we examined the effects of scaling a constant per-device workload across multiple hardware world sizes by increasing the number of devices while maintaining a *fixed local batch size*, which results in an increased global batch size as the number of devices scale.

By contrast, we examine the effects of using model parallelism to train workloads with a *fixed global batch size* while varying the hardware world size, which results in decreasing effective local per-device batch sizes as the number of devices increases. This is representative of industry settings where excess compute resources are allocated for a single training run; and there is a desire to minimize the time to complete a training run as opposed to maximizing the hardware utilization.

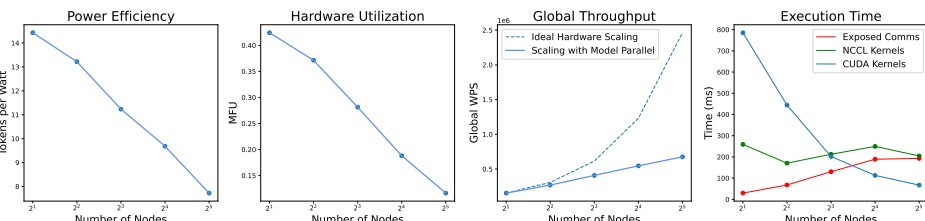

Figure 7: *Training with Fixed Global Batch Size Over Increasing Number of Nodes*. We select the optimal parallelization strategy as determined by the experimental results displayed in Figure 5 for configurations of 2, 4, 8, 16, and 32 H100 nodes to train with global batch size of 32. Even with optimal parallelization strategies, local throughput and hardware utilization declines with world size.

In Figure 7, we show that when training with a fixed global batch of 32 across 2 to 32 nodes – allocation of additional devices yields diminishing returns in global throughput and reduced local hardware utilization. To distribute a fixed workload across more devices, it is necessary to introduce excess degrees of model parallelism which results in insufficient amounts of computation being allocated to each accelerator which we observe as reduced execution time for CUDA kernels. At sufficiently large scales, excess parallelism causes previously compute-bound workloads to become communication bound and reductions in hardware utilization, which we observe over in decreases in MFU of $40\%$ when training with 2 nodes to less than $15\%$ with 32 nodes.

Additionally, we find that these trends persist at pretraining scale with limited marginal returns for increasing the number of hardware accelerators when training both LLAMA-7B and 70B models in Appendix E. We observe that increasing the number of devices from 512 to 2048 GPUs improves global throughput and decreases the per-device MFU local hardware utilization by more than $30\%$.

## 4.4 SCALING THE HARDWARE GENERATION

In Figure 6, we examine the effects of scaling the hardware speed with comparisons between A100 and H100 clusters. In both cases, there exist model parallelism configurations which both increase the overall throughput and reduce the amount of exposed communication relative to data parallel baselines (i.e. total model parallelism equal to one).

When comparing the distributed training performance of the previous generation A100 to the faster H100 hardware when using the optimal parallelization strategy for each platform, the MFU hardware utilization *decreases* from $59.67\%$ to $40.77\%$ The reduction in hardware utilization can be attributed to increases in percent of exposed communication ($+12.83\%$) that emerge due to asymmetric improvements in communication and computation speeds. While improvements are made to both the communication bandwidth and computation speed between the A100 and H100 architectures, the extent to which training is *communication bound increases further with hardware generation* as improvements to computation speed results in shorter computational kernels which increases the difficulty in overlapping hardware which outpaces the rate at which data transfer improves (See Table

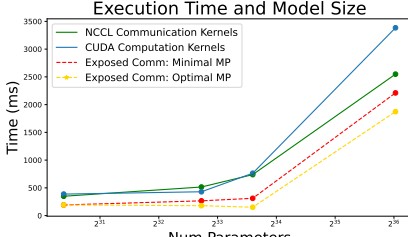

Figure 8: *Communication and Computation Both Scale with Model Size*. As computation load increases with model size, so does the total and exposed communication. At all model scales, model parallelism can be used to reduce exposed communication.

1). In Appendix F, we conduct additional experiments on a V100 cluster in which we similarly find that highest throughput is achieved with model parallelism.

### 4.5 SCALING THE MODEL ARCHITECTURE

We examine the effects of scaling the size of the neural network architectures across 1B, 7B, 13B, and 70B parameters. One might assume that increases in model parameterization solely increases the size of computation while leaving communication unaffected. However, as the number of parameters in a model scale, the volume of communication required for parameter materialization and gradient scattering increases jointly with the size of the computational operations (i.e. matrix operations with larger hidden dimensions). In Figure 8, we consider the optimal model parallelism strategy for each model architecture by sweeping viable tensor and pipeline parallel configurations and observe that the volume of *exposed communication* likewise increases with model size, resulting in lower hardware utilization as models scale.

Additionally, we find that across architecture scales there exist model parallelism strategies beyond the data parallel baseline or the minimal degree of model parallelism (for the 70B parameter model that does not fit on a single GPU) that reduce the volume of exposed communication for all model sizes; and yield higher hardware utilization and throughput.

### 4.6 SCALING THE COMPUTE WORKLOAD

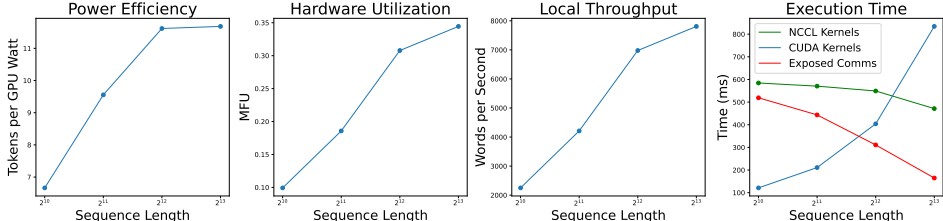

Figure 9: Increased sequence lengths yields larger compute kernels which better overlap with NCCL communication kernels, resulting in *lower exposed communication, higher hardware utilization and power efficiency*.

Finally, we examine the effects of varying the context length in Figure 9. When there is available local GPU memory, increasing the sequence length increases the computational workload allocated to each device without increasing the communication load, yielding improved the throughput, hardware utilization and power efficiency. However, for a fixed world size, reparameterization of the training process in this manner is often not feasible as alterations to the per-batch sequence length affect the training dynamics predicted by computation-architecture scaling laws (Kaplan et al., 2020; Hoffmann et al., 2022a).

## 5 TRENDS IN SCALING AND IMPLICATIONS

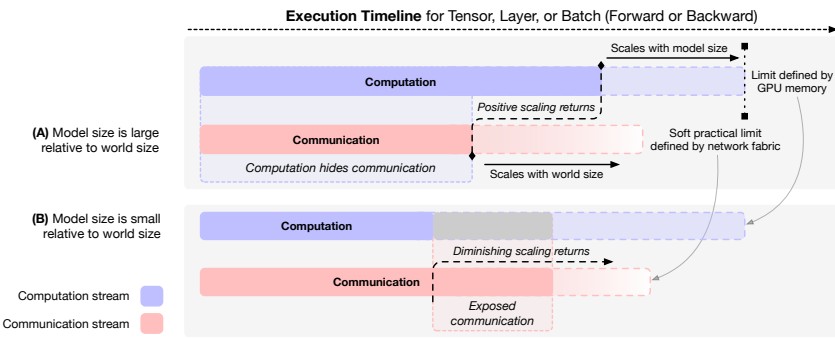

Figure 10: Two distinct training setups and their corresponding concurrent computation and communication streams, executing in parallel. In **(A)**, model size is large relative to world size; computation per-device hides communication cost and scaling the number of devices incurs no cost. In **(B)**, model size is small relative to world size. Communication is not hidden by computation and is exposed; scaling of world size incurs overhead and gives poor marginal gains in training throughput.

**Not all FLOPs are equal.** Existing compute-optimal scaling laws (Hoffmann et al., 2022a; Tay et al., 2023; Dehghani et al.) and workload performance measures are based predominantly on FLOPs or metrics derived therefrom. These fail to take into consideration underlying massively parallelized distributed hardware which requires communication to execute these workload. Local arithmetic throughput per-accelerator does not translate into end-to-end performance due to bounding factors in other hardware components such as network fabric. Integrating holistic information about hardware into scaling practice is essential given that collective communication dominates execution time at scale; scaling laws should be both *compute and communication optimal*.

**Communication-Computation Dynamics Change at Scale.** In distributed training over large-world sizes, the scaling properties of collective communication primitives leads to increased exposed communication and communication-boundedness – which motivates the use of alternative parallelization strategies beyond traditional data parallelism (see Figures 3, 5, 6). This motivates the need for development of parallelization strategies sensitive to the marginal communication costs of increasing world size.

**Additional scale only marginally improves throughput.** Capability and capacity tradeoffs at scale for a fixed global batch size lead to declining marginal improvements — Figures 4 and 7 show emergent upper bounds in the effectiveness of scale as related to model size. If the pace of increases in model size slows — additional scale will do little to improve throughput given fixed recipes, further removing the incentive to scale up without algorithmic modifications.

**Training one large model is less power-per-token efficient than training many smaller ones.** Given aforementioned ceilings in scale, algorithmic paradigms which train ensembles of multiple smaller models will continue to proliferate, with hardware scaling serving growing the number of models in the ensemble. Communication and computation must jointly improve to alleviate bottlenecks for large model training. The current imbalance in rates of improvement of communication and computation constrains new hardware's utility. Figure 1 demonstrates that while power utilization increases linearly, hardware utilization and global throughput both increase sublinearly.

**Improvements in networking within nodes improves scale-out performance.** Inter-node bandwidth is lower as a result of constraints around network fabrics. While improving fabrics may improve performance, increasing node size – that is, building nodes with more accelerators with fast, local interconnects – also increases the total amount of memory and thus the upper bound for degrees of model parallelism. NVIDIA's GB-200[6] features the first increase in NVLink-connected node-size since the DGX-1 P100 in 2017, from 8 to 72 accelerators, with a total of 1 TB of interconnected GPU memory per node. Speedups in inter-node bandwidth and larger collections of high-speed GPU memory will alleviate communication boundedness at large scales.

**Performance benchmarking fails to extrapolate across scales and hardware generations.** As a result of how collective communication primitives for modern parallelism strategies scale, conventional metrics for measuring performance in distributed settings, such as total FLOPS or throughput on smaller scale systems, cannot be extrapolated from small to large-scale without properly accounting for communication dynamics.

## 6  CONCLUSION

In this work, we examine the effects of scaling: parallelization strategies, model architectures, and hardware platforms. We show that communication boundedness worsens at scale and with newer hardware generations, and are persistent across model sizes. Additionally, we show that these trends lead to the emergence of viable parallelism alternatives for distributing deep learning training workloads. Finally, we show that these trends culminate in significant diminishing returns on training performance with respect to real-world resources of power and throughput.

---

[6]NVIDIA GB-200 Datasheet

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

## A RELATED WORK

**Performance Analysis of Deep Learning Systems.** Deep learning poses a unique workload different from traditional high-performance computing settings – with complexity arising from: memory availability and hierarchy; and variable speeds of computation and communication. Prior research has explored the performance properties of individual accelerators (Wang et al., 2020),common workloads (Hsia et al., 2023; Ardalani et al., 2024), and efficient methods for maximizing hardware utilization of these workloads (Hagemann et al., 2023).

Concurrently, several benchmarks have been developed to provide canonical workloads and incentivize efficiency improvements (Mattson et al., 2020; Reddi et al., 2020; Peng et al., 2023). These evaluation suites often only measure the overall end-to-end system performance of standard training and inference recipes (i.e. throughput or wallclock training time) Williams et al. (2009); abstracting away the underlying system. In our work, we seek to examine the joint scaling effects on these downstream performance and system-level utilization metrics as we vary these components (i.e. hardware, model architecture, and parallelization) together.

**Scaling Properties of Deep Learning.** Previous work investigating the scaling properties of neural network training has largely studied the effects of varying the data volume, training compute budget, and model architecture (Hoffmann et al., 2022a; Kaplan et al., 2020; Tay et al., 2023; Porian et al., 2024). These works primarily examine the impact of these factors on the pretraining loss and downstream finetuning performance of the model with respect to the theoretical amount of computational resources allocated (i.e. number of FLOPs).

However, these analyses assume that workload performance scales directly with the amount of computation regardless of the underlying hardware platform and frameworks. In practice, theoretical measures (i.e. FLOPs) are known to be imprecise representations of end-to-end real-world performance (e.g. latency, throughput) due to performance bounds that emerge from management of the computational graph, data transfer, and communication bottlenecks (Dehghani et al.; Fernandez et al.) – or as we show due to communication boundedness.

Additionally, as the scale of deep learning systems has grown, their efficiency has emerged as a serious concern with commensurate scaling in the environmental, financial, and computational resources required to execute such workloads (Wu et al., 2022; Schwartz et al., 2020; Patterson et al., 2022; Luccioni et al., 2024a;b; Strubell et al., 2019; Wu et al., 2024).

## B LIMITATIONS AND FUTURE WORK

In this work, we consider a set of common data and model parallelization techniques for distributing training of neural networks. However, there are additional methods for workload parallelization and memory footprint reduction such as DeepSpeed Zero (Rasley et al., 2020), parallelization of loss computation, and other forms of optimized kernel implementations.

In our investigation across computing platforms, we primarily consider variations in the speed of compute (i.e. GPU generation). In future work, we plan to demonstrate the consistency of the observed trends across settings with variable speeds of communication (i.e. varying speed of internode fabric by comparing InfiniBand interconnects with common alternatives such as RDMA over Converged Ethernet, RoCE).

Additionally, our work is focuses on the training of neural networks based on the Transformer neural network architecture and GPU hardware accelerators. Although we expect our findings to be consistent across other model architectures and hardware platforms, we reserve that examination as areas for future work. Likewise, we focus our investigations on GPUs as it is the most commonly used and easily available hardware accelerator. We expect that similar trends and tradeoffs between communication and computation would occur for alternative hardware accelerator architectures such as TPUs, IPUs, etc.; however we leave exploration of these settings for future study.

## C SOFTWARE AND HARDWARE DETAILS

Training is conducted in `bfloat16` precision with a Megatron-inspired framework and further optimizations provided by FlashAttention-2 (Dao) and xFormers (Lefaudeux et al., 2022). For our primary experiments, we trained models using PyTorch 2.3.1 built with CUDA 12.1, with attention implementation provided by XFormers 0.27.

In supplementary experiments with V100 GPUs in Appendix F, models are trained in `fp16` with loss rescaling and CUTLASS (Thakkar et al., 2023) attention kernels on Volta hardware – due to limited hardware support on older Volta hardware. Nodes within the V100 cluster consist of 8-GPU setups connected with first-generation NVLink in a Hybrid Cube Mesh (HCM) topology.

We compute the runtime of communication and computation kernels by using PerfettoSQL to query Kineto profiles extracted by the PyTorch profiler, which is built on top of NVidia CUPTI to identify relevant NCCL and CUDA kernels, respectively. containing both the CPU and CUDA operations. In Table 1, we provide additional details on the hardware platforms used for running our experiments.

|  | V100 [7] | A100 [8] | H100 [9] |
|---|---|---|---|
| Tensor Core BF16 FLOPS | 125 TFLOPS | 312 TFLOPS | 990 TFLOPS |
| GPU HBM | 900 GB/s | 2 TB/s | 3.35 TB/s |
| NVLink (GPU to GPU Comm) | 300 GB/s | 600 GB/s | 900 GB/s |
| Internode InfiniBand (Node to Node) | 100 GB/s | 200 GB/s | 400 GB/s |

Table 1: Nvidia Reported DGX-Node Specifications by Generation.

## D ADDITIONAL EXPERIMENTS: MODEL PARALLELISM IN ALTERNATE SETTINGS

We extend the experiments from Section 4.2, in which we examine the effectiveness of model parallelism via Tensor and Pipeline parallelism across other hardware settings and computational workloads. In the analysis in §4.2, we consider the setting in which LLama-7B is being trained on 32 DGX H100-80GB nodes with a batch size of 2 – yielding relatively high hardware utilization (MFU) and memory utilization (¿60GB).

Additionally, we consider the effects of model parallelism in settings with lower hardware utilization, due to either: (1) smaller per-device workloads as determined by reduced effective local batch sizes (Figure 11a); or (2) larger communication loads from training in a increasingly distributed hardware settings (Figure 11b). In both regimes, there are a larger number of viable model parallelism strategies.

---

[7]NVIDIA DGX-1 (V100) Whitepaper

[8]NVIDIA DGX A100 Whitepaper

[9]NVIDIA DGX H100 Whitepaper

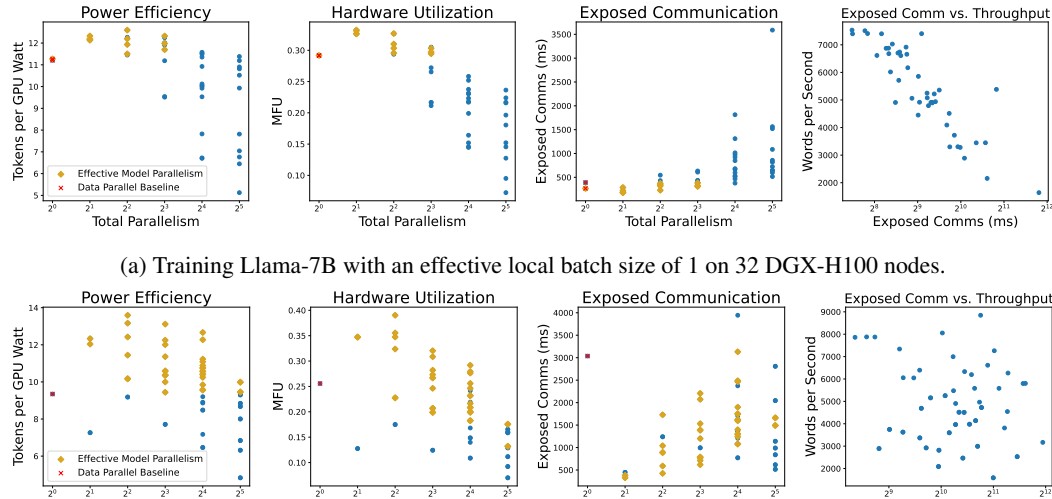

(a) Training Llama-7B with an effective local batch size of 1 on 32 DGX-H100 nodes.

(b) Training Llama-7B with an effective local batch size of 2 on 256 DGX-H100 nodes.

Figure 11: In regimes that are low in arithmetic intensity or communication bounded, there exist many viable strategies for model parallelism that: alleviate communication boundedness, increase power efficiency and hardware utilization.

# E ADDITIONAL EXPERIMENTS: FIXED GLOBAL BATCH SIZE AT PRETRAINING SCALE

We extend the experiments from Section 4, in which we increase the allocation of hardware accelerators to a fixed computational workload with a constant global batch size – i.e. increasing the degree of parallelism across more accelerators without increasing the local effective batch size.

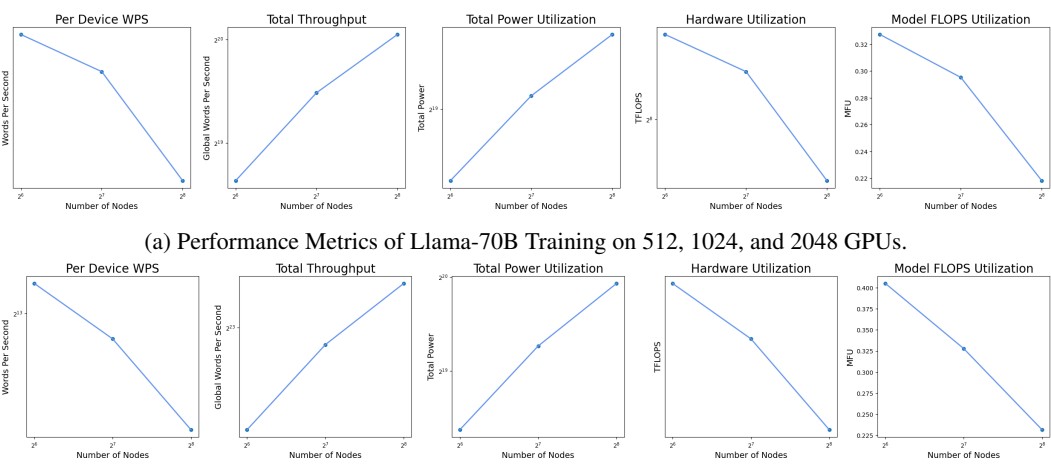

(a) Performance Metrics of Llama-70B Training on 512, 1024, and 2048 GPUs.

(b) Performance Metrics of Llama-7B Training on 512, 1024, and 2048 GPUs.

Figure 12: At pretraining scale, both Llama-7B and 70B observe regressions in hardware utilization and per-device local throughput as the number of devices is increased for a fixed computational workload.

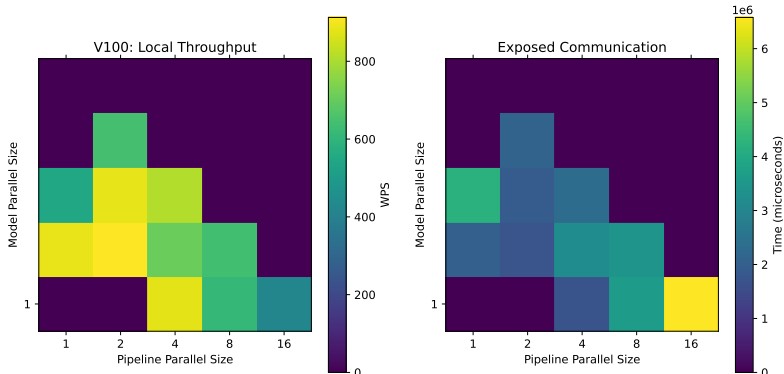

Figure 13: Throughput and Exposed Communication for Model Parallelization Strategies on V100.

## F  ADDITIONAL EXPERIMENTS: PREVIOUS GENERATION V100 HARDWARE

In addition to our primary experiments in Section 4.2, we conduct additional experiments using older V100 GPUs from the Volta architecture training a Llama-7B model with an effective local batch size of 1 on 32 nodes. We observe similar trends in which small degrees of model parallelism improve overall throughput at scale. However, due to lack of additional optimized kernels (e.g. CUTLASS vs FlashAttention kernels) and Ampere hardware optimizations, we observe that the transition to Ampere A100 GPUs in fact improves overall hardware utilization.

## G  EFFECTS OF SCALING WORLD SIZE ON MEMORY UTILIZATION

In fully-sharded data parallelism (FSDP), increasing the number of data parallel instances decreases the memory utilization per-GPU as parameters and gradients are sharded over additional data parallel instances. However, the memory savings diminish with device world size.

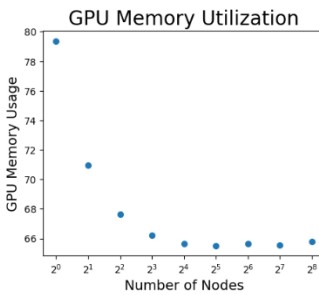

Figure 14: Increasing the data parallel world size reduces local per-GPU memory utilization, but reductions diminish with scale.

