# OpenReview forum: "Hardware Scaling Trends and Diminishing Returns in Large-Scale Distributed Training"
_ICLR.cc/2025/Conference — ICLR 2025 Conference Withdrawn Submission_

### Official Review · Reviewer_Xci8 · 2024-10-29

**Soundness:** 2
**Presentation:** 3
**Contribution:** 2
**Rating:** 3
**Confidence:** 4

**Summary:**

The authors performed extensive experiments on hardware utilizations (MFU, tokens/w, etc) when number of accelerators (e.g., GPUs) scales to hundres of GPU server nodes (thousands of GPUs) and found:
1. Combination of different parallelism, e.g., FSDP and TP/PP can leverage the heterogeneous intra/inter-node interconnect BW more efficiently than FSDP along
2. Yet the system becomes communication (latency or bw) bound and has diminished return of MFU beyond certain scale

**Strengths:**

+ Extensive experiments on most popular 3D parallelims on modern models/hardwares at large scale (thousands of GPUs)
+ Real world evaluation metrics other than MFU, e.g,. tokens per watt etc
+ Could serve as a quantative handbook for researchers/engineers optimizing distribributed system

**Weaknesses:**

- The 2nd conclusion from the study is well know, i.e, distributed system is typically communication bound. The authors touch a bit on all-reduce could be BW bound while reduce-scatter and all-gather can be latency bound, e.g., Figure 3, but didn't go further provide any quantitative/analytical model or techical details on why the latter two are latency bound

- The 1st conclusion regarding parallelism strategies (and a mitigation to point above) is also relatively well known, e..g, the 3D (FSDP/TP/PP) is quite standard in modern LLM training, and their design and relation to network topogies (NVLink vs Infiniband) is also well studied, e.g., in Megatron series of papers.

- While it is true that distributed system total throughput scales sublinearly at scale, the "scale" the auhors begin to see diminsing returns (e.g., 2000 GPUs, both 7B/70B) are far behind industry leading results, e.g., Llama 3 training can reach 40% MFU at 20, 000 GPUs scale (https://arxiv.org/pdf/2407.21783)

**Questions:**

1. why is there a sudden jump in CCL time from 128 to 256 nodes in last plot of Figure 4?
2. what are the yellow and blue dots in Figure 5? My guess is one is PP and the other is TP.
3. what's the training framework used? Megatron/Deepspeed or in-house developed, how does it compare to these baselines?

---

> ### Author Response · Authors · 2024-11-27
>
> We agree with the reviewer that our work provides insights into other measures of performance beyond standard MFU metrics in realistic training settings – and that the guidance can be useful for practitioners designing large scale distributed training systems.
>
> Below, we address the reviewers concerns with our work. Please let us know if there are additional concerns or clarifications we can provide.
> 1. **Distributed system is typically communication bound.** Although the conclusion that communications emerge at scale is a known property of distributed computing, we demonstrate common regimes in which communication overhead dominates: e.g. the exact regimes in which over-decomposition of a 7B and 70B models on commodity hardware results in communication exceeding compute (Fig 4). \
> \
> Additionally, we provide experimental comparisons showing that these behaviors have worsened due to asymmetric improvements in compute and communication technology across Ampere and Hopper GPU clusters (Sec 4.4). Furthermore, we show that different choices in parallelization strategies can reduce exposed communication beyond the assumed optimal data parallelism baseline (Fig 6).
>
> 2. **Comparisons with Megatron Studies.** Prior work [1,2] studying 3D parallelism with Megatron and similar frameworks does so *without* FSDP or Zero memory optimizations. These works conclude that model parallelism introduces frequent blocking communications which leads data paralellism to be preferred for its non-blocking communications.
> Additionally, Narayanan et al 2021 concludes that Megatron 3D parallelism outperforms Zero-3 without any model parallelism, and speculates that model parallelism may be able to improve utilization of Zero based methods but do not provide experimental results.\
> \
> In our work, we consider training with 3D parallelism and FSDP, commonly used techniques for alleviating memory pressure (e.g. Meta Llama 3.1, ByteDance MegaScale, IBM Granite 3.0, AI2 Olmo) [3,4,5,6]. In this setting, model parallelism is not needed as parameter and optimizer state sharding sufficiently reduces memory pressure – yet we show introduction of _model parallelism is still in fact preferred to data parallelism_ due to reduced exposed communications, contradicting prior work.
>
> 3. **The "scale" the authors begin to see diminishing returns are far behind industry leading results.**
> In our work we show that the diminishing returns emerge in training regimes as few as 2000 GPUs, with communication overhead becoming increasingly an issue at larger sizes. In fact, without adequate parallelization strategies diminishing returns when training on tens of thousands of GPUs becomes an even greater problem.
>
> 4. **Why is there a sudden jump in CCL time from 128 to 256 nodes in last plot of Figure 4?**
> The large increase in collective communication times corresponds to the increases in collective communication costs that are incurred when performing collectives over more devices. As seen in Figure 3, the cost of collective communications increase with number of devices which is seen for all number of nodes. The magnitude of the observed jump between 128 and 256 nodes in Fig 4, is especially noticeable due to the log-x axis and linear y-axis.
>
> 5. **What are the yellow and blue dots in Figure 5? My guess is one is PP and the other is TP.**
> This corresponds to the best parallelism strategy for a certain degree of total parallelism.  We have added labels to the legend for Figure 5 for clarity.
>
> 6. What's the training framework used? Megatron/Deepspeed or in-house developed, how does it compare to these baselines?
> We used an in-house training framework based on Megatron-LM, that has been used to train open models at the scale of hundreds of billions of parameters. The framework in question has achieved comparable MFU and hardware utilization with current SoTA (>38% MFU for 70B models on H100 GPUs, comparable to 35% MFU reported for 76B parameter model with Megatron-LM [2]).
>
> References:
> 1. Narayanan et al., "Efficient Large-Scale Language Model Training on GPU Clusters Using Megatron-LM", Supercomputing 2021
> 2. Hagemann, Johannes, et al. "Efficient Parallelization Layouts for Large-Scale Distributed Model Training." Workshop on Advancing Neural Network Training: Computational Efficiency, Scalability, and Resource Optimization (WANT@ NeurIPS 2023). 2023.
> 3. Dubey, Abhimanyu, et al. "The llama 3 herd of models." arXiv preprint arXiv:2407.21783 (2024).
> 4. Jiang, Ziheng, et al. "{MegaScale}: Scaling large language model training to more than 10,000 {GPUs}." 21st USENIX Symposium on Networked Systems Design and Implementation (NSDI 24). 2024.
> 5. Hess, Peter. “Training AI Models Faster than Ever.” IBM Research, IBM, 18 Sept. 2024, research.ibm.com/blog/pytorch-2024-training-models. Accessed 27 Nov. 2024.
> 6. Groeneveld, Dirk, et al. "Olmo: Accelerating the science of language models." arXiv preprint arXiv:2402.00838 (2024).

---

> > ### Comment · Reviewer_Xci8 · 2024-11-28
> >
> > Thanks for your reply but the author's response didn't address my main concerns.
> >
> > 1 & 3. The authos didn't directly repond to my concerns (e.g., why is all-reduce BW bound and reduce-scatter and all-gather are latency bound while all-reduce are typically achieved with the latter two), it is a reiteration of the points in the manuscript.
> >
> > 2. Comprisons with Megatron studies. Megatron-turing [https://arxiv.org/pdf/2201.11990] has used both FSDP + PP + TP and demonstrated over 30% MFU, over 420 nodes. Addtionally Palm [https://arxiv.org/pdf/2204.02311] has also demonstrated TP + FSDP over 6000+ TPUs with 46% MFU. Llama 3 [https://arxiv.org/pdf/2407.21783] used FSDP + PP + TP + SP/CP over 16K H100s, and achieved 40% MFU. Thus 3D parallelism + FSDP has been exercized extensively and all of above works have demonstrated higher MFU than this work.
> >
> > 4. Sudden jump. The authors again just reiterate over well-known facts. And the explanation of plot scale does not make sense, even if the CCL is fully latency bound, from 128 to 256 nodes there should be only a 2X jump, yet there is 4X jump (1000 -> 4000ms), how is this explained by plotting?
> >
> > 6. I didn't find any systematic comparison to Megatron LM in the updated version to support the author's claim.
> >
> > Thus I am maintaining my rating.

---

### Official Review · Reviewer_7QXa · 2024-10-30

**Soundness:** 4
**Presentation:** 4
**Contribution:** 4
**Rating:** 6
**Confidence:** 4

**Summary:**

The authors conduct a large-scale empirical study on large-scale LLM training workloads and presents the following insights:

1. As the total number of accelerators used for training large models increases, there are diminishing returns in performance. This means that even with optimized hardware and parallelization strategies, adding more accelerators yields progressively smaller improvements in throughput and efficiency.

2. The overhead incurred from certain distributed communication strategies can lead to scenarios where parallelization methods previously considered sub-optimal become preferable. This highlights the critical role of communication efficiency in large-scale training.

3. The study characterizes the scaling properties of sharded training, demonstrating that the efficiency of distributed training is heavily influenced by the balance between computation and communication costs. As model sizes and batch sizes increase, the communication volume also rises, making the training process increasingly communication-bound rather than compute-bound.

4. The experiments show that hardware utilization and throughput regress as the number of devices increases for a fixed computational workload. This indicates that scaling up the number of accelerators does not guarantee improved performance.

**Strengths:**

The thorough empirical study bridges the gap between existing scaling laws that connect loss to training FLOPs and real-world training configuration decision-making.

The insights presented in the paper answer many open questions in training configuration selection where no prior work has been done systematically studied on.

**Weaknesses:**

1. "Training one large model is less power-per-token efficient than training many smaller ones." The part on energy efficiency seems a bit out of place. If there is a need/workload to train a large model, then training many smaller ones would not replace the need to train a larger one. I am not sure if this takeaway fits into the theme of the rest of the paper, which answers the important question of how different parallelism configurations affect training throughput.

2. While I understand this may not be feasible from a resource perspective, knobs such as global batchsize/sequence length would affect loss/downstream accuracy, which leads to a new space of tradeoffs. This regime is unexplored and may impact the quality of the trained model. Some results in this line would further strengthen the paper, such as if you pick the parallelism configuration you find to be most efficient, how does the training loss curve differ from existing known configurations used to train other open models of the same size? Or how and why existing models have been using suboptimal configurations during training.

3. "Additional scale only marginally improves throughput." The author argues that with fixed global batchsize, scaling has marginal returns. However, this seems to be an unrealistic setting. Using more nodes serves the exact purpose of accommodating larger global batchsize on many occasions with the assumption that using larger batchsize for LLM pretraining has minimal/acceptable impact on downstream performance/loss.

**Questions:**

My concerns are raised above. If the authors can address my concerns above, I would be happy to raise my rating.

---

> ### Author Response · Authors · 2024-11-24
>
> We agree with the reviewer that our work provides an empirical handbook and guidance for reasoning about the dynamics between training configurations and underlying hardware – and that our work addresses gaps in the literature grounding existing scaling trends in practical implementations.
>
> We address the reviewers comments below.
> 1. **Single Large Model vs Multiple Small Models.** As the reviewer notes, multiple models do not address the need for a model of a given size in the case where there is a single large model desired. However, it is common practice to train and iterate on proposed methods with smaller scale models and leverage neural scaling laws [1,2]. We show that this is not only faster to train due to fewer required training FLOPS, but also faster in practice and more power efficient due to increased hardware utilization.  Additionally, this observation is useful as guidance on the organizational perspective to allocating compute fleet-wide in which it is important to ensure high hardware utilization of resources.
>
> 2. **Hardware and Performance Scaling Laws.** As mentioned in our first recommendation in Sec 5, we agree that there remains an open question as to unifying the empirical scaling laws that govern model performance (i.e. FLOPS x Training Data x Loss). We believe that our work is a first step towards such a unified performance model as we demonstrate some of the considerations that need to be accounted for prior to mapping real-world efficiency metrics to theoretical FLOPS.
>
> 3. **Marginal Returns from Scale.** It is true that the number of nodes are often scaled to accommodate larger models and global batch sizes. However, in settings – such as in industry –  with a surplus of computing resources, it is the case that additional compute resources can be allocated to speedup the runtime of a single training job.  We show that even with additional resources in these settings, the communication overhead limits  the observed speedup despite large  increases in GPU-hours and power consumption.
>
> References:
> 1. Hoffmann, Jordan, et al. "Training compute-optimal large language models." arXiv preprint arXiv:2203.15556 (2022).
> 2. Kaplan, Jared, et al. "Scaling laws for neural language models." arXiv preprint arXiv:2001.08361 (2020).

---

> > ### Comment · Reviewer_7QXa · 2024-11-25
> >
> > I appreciate the author's response and am maintaining my score.

---

### Official Review · Reviewer_UQA7 · 2024-11-02

**Soundness:** 1
**Presentation:** 1
**Contribution:** 1
**Rating:** 3
**Confidence:** 4

**Summary:**

The authors address three questions in this paper：
1. How do hardware scaling and parallelization strategies affect the efficiency of large-scale model training?
2. What’s the impact of different parallelism methods(data, tensor, pipeline) on training performance?
3. How do communication costs limit scalability, and when do we see diminishing returns?

To answer these questions, the authors measured performance metrics, including throughput (words per second, tokens per second), power efficiency, and communication overhead (time and exposed communication) using the Llama 2 model families (1B and 70B) across different generations of NVIDIA GPUs (V100, A100, and H100) with combined parallelization strategies of data-, tensor-, and pipeline-parallelism. These measurements are from a few to 2,048 GPUs.

The experiments compare the scalability of data, tensor, and pipeline parallelism, evaluating the impact of different hardware configurations and GPU counts on training speed and efficiency. Additionally, the authors examined how communication overhead constrains scalability as system size increases.

Based on the performance measurements, the authors draw three conclusions:
1. Communication limits scaling: as hardware scales up, performance gains slow down due to communication overhead.
2. Model parallelism helps reduce some overhead but has limitations, especially across multiple nodes.
3. Increasing the total number of accelerators for large model training quickly leads to diminishing returns, even with optimized hardware and parallelization strategies, indicating poor marginal performance per additional unit of power or GPU-hour.

The authors conclude the paper with five trends in scaling and implications in Section 5.

**Strengths:**

1. It is promising to use real-world LLMs for performance measurements
2. The testbed with 2,048 GPUs is sufficient to justify the scaling trends.

**Weaknesses:**

None of the scaling trends in Section 5 is new or unexpected.
1. Not All FLOPs Are Equal: The authors point out that existing compute-optimal scaling laws and performance metrics primarily rely on FLOPs or derived metrics, which fail to take into consideration underlying massively parallelized distributed hardware which requires communication to execute these workload.

This statement overlooks the latest LLaMA 3.1 paper (https://arxiv.org/abs/2407.21783), which considers the impact of communication in addition to FLOPs-based metrics.

2. Communication-Computation Dynamics Change at Scale. The authors claim that it is the low scalability of the collective communication primitives that motivates alternative parallelization strategies. This statement is incorrect, as the tensor-, pipeline-, and context-parallelism are motivated by the need of higher learning capability in LLMs.

3. Additional scale only marginally improves throughput. The paper overlooks the opportunity of context-parallelism, which will be significantly improved with additional computing resources. This idea has been demonstrated in Llama 3.1 paper, where Meta trains the 405B model with 16,384 H100 GPUs using ring-attention.

4. Training one large model is less power-per-token efficient than training many smaller ones. This is obvious as smaller models have less communication requirement, which results in better computation efficiency and, thus, power-per-token efficiency. This is not a new insight.

5. Improvements in networking within nodes improves scale-out performance. This statement is misleading, as an enhanced intra-node network would improve the overall training performance. However, the contribution of intra-node networks is independent of inter-node communication.

6. Performance benchmarking fails to extrapolate across scales and hardware generations: The paper concludes that performance benchmarks cannot be extrapolated across different scales and hardware generations, but the experiments presented do not provide sufficient evidence to support this claim.

There exists ambiguity in Scaling Model Parallelism (Section 4.2): When scaling model parallelism, the degree of model parallelism is described as the product of tensor and pipeline parallelism, however, the mapping between a model-parallelism value to the tuple of tensor- and pipeline-parallelism is not unique. It is unclear if the authors examine all possible combinations of tensor- and pipeline-parallelism for a given model-parallelism setting. For example, the total degree of 16 can be 2-way tensor-parallel and 8-stage pipeline, or 4-way tensor-parallel and 4-stage pipeline. This ambiguity makes it unclear how the scaling experiment was conducted.

**Questions:**

Please address all questions in the Weakness section.

---

> ### Author Response · Authors · 2024-11-24
>
> While we acknowledge the reviewer’s concerns around the alignment of demonstrated trends with expectations.  In contrast, to the Llama 3.1 technical report which mentions memory consumption and performance projection tools used to explore parallelization strategies in limited detail, our work provides empirical results that demonstrates **scales and regimes** in which different bottlenecks and overhead dominate performance.  This work can assist in characterization of both the computational workloads and systems in modern training environments.
>
> For example, we show that total execution time of collective communications exceeds that of computation which results in unavoidable exposed communication, when training 7B models on >128 H100 GPUs (Fig 4); and that excess degrees of model parallelism can similarly result in unavoidable exposed communication (Fig 5).

---

> > ### Author Response · Authors · 2024-11-24
> >
> > Below, we address specific reviewer concerns with our work.
> > 1. **Communication Impact on Scaling Laws.** While the Llama 3.1 paper discusses the impact of communication and parallelism strategies, this reference is a technical report that does provide details or experimentation studying the dynamics between different parallelism strategies. In our work, we conduct extensive empirical experimentation showing the impact of these parallelisms on communication and overall performance. \
> > \
> >   In describing compute-optimal scaling laws, the Llama 3.1 technical report solely relies on FLOPS and Training Tokens (Fig 2 and 3 of the Llama 3.1 report) – our recommendation in 5.1 is that these scaling laws are best grounded based on real world measurements (such as wall clock time and GPU-hrs) which are communication dependent.
> >
> > 2. **Communication-Computation Dynamics Change at Scale.** Could the reviewer please clarify as to the relationship between learning capabilities and the need for parallelization strategies?\
> > \
> > Parallelization strategies such as Tensor, Context, and Pipeline are primarily developed to distribute a fixed computational workload over a hardware platform not due to training dynamics or learning capabilities. The motivation for the development of different strategies and schedules is largely to address the communication dynamics for data of varying sizes, blocking operations, and network topology [1,2,3].
> > 3. **Marginal Returns at Scale and Context Parallelism.** Diminishing returns at scale is a conclusion regardless occurs for all 4D parallelisms (tensor, pipeline, context, data) as communication costs scale and become unavoidable, which we demonstrate via experimentation across the commonly used 3D parallelisms and variable hardware platform sizes – and in Appendix D, we include analysis comparing Tensor Parallelism with Context Parallelism and show that Context Parallelism does not alleviate this issue.
> > 4. **The tradeoff between training multiple small models and a single large model.** While a natural conclusion, the statement provides guidance for maximization of hardware utilization.  It is common practice to train and iterate on proposed methods with smaller scale models and leverage neural scaling laws [4,5]. We show that this is not only faster to train due to fewer required training FLOPS, but also faster in practice and more power efficient due to increased hardware utilization.
> > 5. **On improvements to Intranode Networking.** In the referenced section, we suggest that improvements in intranode networking that *increase* node size can alleviate communication boundedness by allowing for parallelization strategies which rely on shifting more communication to within the highspeed local interconnects as opposed to internode fabric.
> > 6. **Scaling Across Hardware Generations.** In Section 4.4 and 4.5, we study the impact of training on GPUs from the Ampere and Hopper generations. We show that due to asymmetric increases in communication and computation speeds across generations in which the internode fabric speeds have increased at a slower rate than compute speeds, the expected hardware utilization has decreased with advances in GPU architecture.\
> > \
> > Similarly in Sec 4.1 and 4.3, we show that performance behaviors are not constant across **scales of hardware** and that benchmarking at small scales (compute bound) does not generalize to large scale (communication bound).
> >
> > In regards to exact model parallelism configurations, we conduct a sweep over products of tensor, pipeline, and context parallelism degrees in [1, 2, 4, 8 ,16] – as described in Sec 3: Parallelization Strategies.
> >
> > References
> > 1. Narayanan, Deepak, et al. "PipeDream: Generalized pipeline parallelism for DNN training." Proceedings of the 27th ACM symposium on operating systems principles. 2019.
> > 2. Shoeybi, Mohammad, et al. "Megatron-lm: Training multi-billion parameter language models using model parallelism." arXiv preprint arXiv:1909.08053 (2019).
> > 3. Huang, Yanping, et al. "Gpipe: Efficient training of giant neural networks using pipeline parallelism." Advances in neural information processing systems 32 (2019).
> > 4. Hoffmann, Jordan, et al. "Training compute-optimal large language models." arXiv preprint arXiv:2203.15556 (2022).
> > 5. Kaplan, Jared, et al. "Scaling laws for neural language models." arXiv preprint arXiv:2001.08361 (2020).

---

> > > ### Comment · Reviewer_UQA7 · 2024-11-25
> > >
> > > I appreciate the authors' comprehensive reply. I would encourage the authors to think about the claims from another way. If the scaling trends in this manuscript are true, how do we interpret Meta's Llama 3.1 405B model? It was trained on 16,384 H100 GPUs. Llama 3.1 405B achieves better downstream task performance than smaller models. It uses 8X more GPUs for noticeable improvements. It exploits context parallelism for higher learning capability.

---

> > ### Comment · Reviewer_UQA7 · 2024-11-24
> >
> > Llama 2 7B is with 4K sequence length and 4M token global batch size. Distributing 4M/4K=1024 sequences over 128 H100s leads to 8 sequences per H100. What's the impact on individual GPU utilization? It is difficult to justify the stated finding without such performance profiling.

---

> > > ### Author Response · Authors · 2024-11-28
> > >
> > > While the exact pretraining setup mentioned is not possible, as an effective local batch size of 8 for Llama 7B with 4096 tokens does not fit on a single GPU without using CPU offloading or activation recomputation. We do consider realistic pretraining settings in Figure 4 and in Appendix E.
> > > * In Figure 4, we perform strong scaling experiments in which models are trained with a _fixed local batch size_ of two using between 8 and 2048 GPUs, equivalently 128 and 256 DGX nodes -- spanning global batch sizes of 16 to 2048 examples. At these settings, the effects of communication overhead grow with scale as observed in reduced throughput and utilization yielding lower GPU utilization.
> > > * In Appendix E, we train models with the fixed global batch size of 1024 and show that introduction of additional accelerators beyond the minimum required improves global throughput but at reduced efficiency yielding lower GPU utilization.

---

> > > > ### Comment · Reviewer_UQA7 · 2024-12-02
> > > >
> > > > "a fixed local batch size of two using between 8 and 2048 GPUs, equivalently 128 and 256 DGX nodes -- spanning global batch sizes of 16 to 2048 examples". This is changing the batch size. It is weak scaling.
> > > >
> > > > The hardware utilization figures in Figure 6 are unreadable without data labels.
> > > >
> > > > Now let's look at the claimed scaling trend: "Additional scale only marginally improves throughput."
> > > > The Llama 3.1 paper (Table 4) shows 43% MFU for 405B on 8,192 GPUs.  While with 16,384 GPUs, which is 2X as the previous case, the MFU is 41%. That means, the throughput increases almost 2X from 8,192 to 16,384 GPUs.
> > > > This clearly proves that the scaling trend claimed in the paper is false.

---

### Official Review · Reviewer_An5A · 2024-11-04

**Soundness:** 3
**Presentation:** 3
**Contribution:** 3
**Rating:** 5
**Confidence:** 5

**Summary:**

The authors of this paper stated that recent advancements in neural network capabilities, particularly in large language models (LLMs), are largely driven by increases in model size, training data, and computational resources. Training these expansive networks often involves distributing tasks across thousands of hardware accelerators like GPUs, necessitating effective orchestration of computation and communication across large clusters. This study highlights the importance of hardware configuration and parallelization strategies for efficient scaling. Key findings include: (1) certain distributed communication strategies that were previously deemed sub-optimal may become more favorable beyond specific scales due to overheads; and (2) increasing the number of accelerators can lead to diminishing returns in performance, even with optimized strategies, indicating a decline in the marginal benefits of additional computational power.

**Strengths:**

1. This paper provides evidence that model parallelism yields improved global throughput although previous works suggest the opposite.
2. The experiment setups are well-described and well-established in Section 3.
3. This paper brings up some observations suggesting that data parallelism could be less efficient compared to what people previously thought. Model parallelism in large-scale training is actually not causing many efficiency drops in a large-scale situation due to communication overlapping.

**Weaknesses:**

1. There are some minor concerns and suggestions about this paper. I have listed them in the Questions section.
2. There is no discussion on the training framework which could be buggy and thus making the experiment results less convincing.
3. Figure 6 seems to have the wrong labels, it is using pipeline parallel size 16 and model parallel 1 which is impossible.
4. The current trend in finding optimal MP and DP is testing configs one by one, also MP size vs efficiency is supposed to be a bell curve since model size couldn't be infinitely large. DP helps to scale up the training process and MP is designed to fit the larger models. Some DP and MP comparisons in this paper seem not fair since they do not have the same target in the first place. As in Figure 10, MP helps reduce computation per GPU well DP does not (not considering FSDP).

**Questions:**

1. Can you also add a citation to Sequence Parallel in 2.1.1?
2. The legend arrangement in Figure 3 seems confusing. Also, a legend of the second graph in Figure 4 is missing. Figure 5 is not well explained, like the color difference.
3. All the experiments are conducted with a Megatron-inspired framework as mentioned in Appendix E, is there any pros and cons of this framework? Are there any concerns about this framework? Could it affect the experiment results?

---

> ### Author Response · Authors · 2024-11-24
>
> We thank the reviewer for their feedback and agree that our work provides evidence for the efficacy of alternative parallelization strategies in large-scale training, contrary to conventional knowledge -- and demonstrates the trends in performance as relative speeds of computation and communication scale.
>
> Below, we address the reviewer’s concerns from Weaknesses and Questions:
> 1. Training Framework. Experiments were all conducted with the Megatron-LM based framework described in Appendix E.  In training runs at 1024 GPU-scale, the framework in question has achieved comparable MFU and hardware utilization with current SoTA (>38% MFU for 70B models on H100 GPUs, comparable to 35% MFU reported for 76B parameter model with Megatron-LM [1]).
> 2. Clarity of Figures.
>  - For Figure 3, could the reviewer provide additional details as to what we can clarify for legend placement?
>  - In Figure 4, the dashed line for both the second and third graphs corresponds to ideal hardware scaling (i.e. local throughput remaining constant regardless of the number of nodes).
> - In Figure 5, the colored points correspond to the best parallelism strategy for a given degree of total parallelism. In the revised submission, we have instead highlighted points corresponding to  “Effective Model Parallelism Strategies” in which model parallelism yields higher performance than standard data parallelism.
>  - Labels in Figure 6. We thank the reviewer for catching this, this was a typo in the axis label. Model Parallel Size refers to Tensor Parallelism, which we have corrected in a revision to the submission pdf.
> Please let us know if there are any other questions or concern that we can address.
>
> References Cited.
> 1. Hagemann, Johannes, et al. "Efficient Parallelization Layouts for Large-Scale Distributed Model Training." Workshop on Advancing Neural Network Training: Computational Efficiency, Scalability, and Resource Optimization (WANT@ NeurIPS 2023).

---

> > ### Comment · Reviewer_An5A · 2024-11-27
> >
> > I appreciate authors' reply. Is there any comment on the weakness 4? It is the most important concern on my side. Thanks

---

> > > ### Author Response · Authors · 2024-11-28
> > >
> > > We would like to contrast our work with several prior works which study the effectiveness of parallelization strategies with FSDP, a common approach used in LLM training (e.g. Meta Llama 3.1, ByteDance MegaScale, IBM Granite 3.0, AI2 Olmo) -- and find that the beneficial performance of MP is in contradiction to prior work [1,2,3] which studies 3D Megatron parallelization strategies in the absence of  FSDP.
> > >
> > > We appreciate the reviewer's engagement and address Weakness 4 below.
> > >
> > > >   **The current trend in finding optimal MP and DP is testing configs one by one.**
> > >
> > > >  **DP helps to scale up the training process and MP is designed to fit the larger models.**
> > >
> > > In this work, we show that _incrementally introducing tensor parallelism_ from the FSDP data parallel baseline is effective in reducing exposed communication and increasing throughput even when MP is _no longer required_ from a memory requirement perspective (Figure 6). As such we demonstrate that model parallelism is beneficial in settings outside of reducing local computation just to fit larger models.
> > >
> > > > **MP size vs efficiency is supposed to be a bell curve since model size couldn't be infinitely large.**
> > >
> > > As the reviewer notes, introduction of model parallelism is only effective in certain settings, such as when it can alleviate other communication overheads. In Figure 7, we observe that excess scaling model parallelism to additional devices results in over decomposition of the workload and reduced per-GPU computation such that communication dominates (see Execution Time subfigure). While scaling across additional devices requires more parallelism and does reduce utilization, the most performant strategy for a fixed number of devices often may have non-trivial model parallelism (Fig 5 and 11).
> > >
> > > > **Some DP and MP comparisons in this paper seem not fair since they do not have the same target in the first place. As in Figure 10, MP helps reduce computation per GPU well DP does not (not considering FSDP).**
> > >
> > > In our experiments comparing different model parallelism strategies such as in Figure 5 and 6, we control runs such that they all maintain the same global batch size of 512 examples on 256 GPUs regardless of the parallelism strategy. For instance, when comparing the DP baseline (2 examples per GPU) with a Tensor Parallel, Pipeline Parallel of (2, 2), the model parallel setup is trained with a microbatch of 8 examples -- so both setups have the same _effective local batch size per GPU_.
> > > \
> > > Please let us know if there are other concerns we can address or clarify. Thank you.
> > >
> > > **References.**
> > > 1. Narayanan et al., "Efficient Large-Scale Language Model Training on GPU Clusters Using Megatron-LM", Supercomputing 2021
> > > 2. Hagemann, Johannes, et al. "Efficient Parallelization Layouts for Large-Scale Distributed Model Training." Workshop on Advancing Neural Network Training: Computational Efficiency, Scalability, and Resource Optimization (WANT@ NeurIPS 2023). 2023.
> > > 3. Hanindhito, Bagus, Bhavesh Patel, and Lizy K. John. "Bandwidth Characterization of DeepSpeed on Distributed Large Language Model Training." 2024 IEEE International Symposium on Performance Analysis of Systems and Software (ISPASS). IEEE, 2024.

---

### Official Review · Reviewer_Daiz · 2024-11-05

**Soundness:** 2
**Presentation:** 4
**Contribution:** 2
**Rating:** 3
**Confidence:** 4

**Summary:**

The paper provides an empirical analysis of the performance of training LLMs with a variety of different parallelization configurations. The study includes data- and model-parallelism across a variety of GPU scales, as well as different GPU generations (A100 vs H100) and model sizes. A key takeaway is that many parallelization techniques become communication-bound at scale, and given that compute performance is rapidly outstripping communication performance, this may necessitate rethinking training paradigms for models.

**Strengths:**

1. Large-scale training is becoming increasingly common and having good studies and guidelines for this available to the community is valuable.
2. The paper studies a number of situations and identifies limitations in commonly-used parallelization schemes at scale.
3. The paper is clear and well-written, and its points are easily understood.

**Weaknesses:**

1. The paper's key point is that scaling a fixed model across more accelerators leads to rapidly diminishing returns due to communication overheads. Yet it seems to me that this is just recapitulating the well-known challenges of strong scaling and over-decomposition (_any_ fixed problem becomes communication-bound when it is scaled, unless it is embarrassingly parallel). Indeed, the scientific and high-performance computing communities have long recognized this and have typically preferred weak-scaling paradigms where possible (see, e.g., Gustafson's Law; the classic citation being Gustafson, "Reevaluating Amdahl's Law", CACM 1988). It is not clear to me that the paper is making any real contribution here; perhaps incorporating weak scaling studies would benefit the paper.
2. Some of the insights in the paper in terms of configuration for parallelization strategies (tensor vs pipeline vs data parallelism, etc.) seem to reprise the advice given in Narayanan et al., "Efficient Large-Scale Language Model Training on GPU Clusters Using Megatron-LM", Supercomputing 2021. How does the paper compare with Narayanan et al., and does it draw different conclusions? (Note there are some configurations, e.g., sequence parallelism, considered here but not in Narayanan et al.)
3. Related to the above two points, it seems that many of the configurations the paper evaluates are not actually realistic. For example, due to the communication requirements, FSDP is typically not performed over many nodes as in Figure 4 and instead "hybrid shard" versions are employed where the model is sharded only over a small number of GPUs (e.g., one node) and standard data-parallelism is employed across shards. A similar strategy is used with tensor parallelism. It is thus not clear what the practical value of many of the configurations evaluated is, considering one would not use them in practice as they over-decompose.

These are less critical points, but seem to indicate some underlying issues with the paper:

4. In Section 2.2, the paper states that ring allreduces are "bandwidth-balanced" (I assume this means optimal?) whereas tree allreduces have suboptimal bandwidth but better (logarithmic) latency. This is not fully accurate. It is indeed the case that a tree-based recursive doubling algorithm for allreduce takes time (ignoring computation; $p$ processors, message size $n$, latency $\alpha$, inverse bandwidth $\beta$) $\log p \alpha + n \log p \beta$ and a ring allreduce takes time $2 (p-1) \alpha + 2 \frac{p-1}{p} n \beta$. However, the classic Rabenseifner algorithm for allreduce is tree-based (recursive halving/doubling) and bandwidth-optimal, with time $2\log p \alpha + 2 \frac{p-1}{p} n \beta$ (although in practice ring-based algorithms can perform better depending on the network topology/configuration). See, e.g., Thakur et al., "Optimization of Collective Communication Operations in MPICH", IJHPC 2005; Chan et al., "Collective communication: theory, practice, and experience", Concurrency and Computation: Practice and Experience, 2007.
5. Also in Section 2.2., the paper states that "AllGather and ReduceScatter can only use ring algorithms". This is incorrect: tree-based algorithms (with logarithmic latency terms) are known. See the recursive-doubling or Bruck algorithms for allgather and the recursive-halving algorithm for reduce-scatter. (See above references for details.)
6. In Section 2.1.1, when discussing data parallelism, the paper states that the gather operations for FSDP delay computation. While it is true that FSDP introduces communication on the critical path, the situation is not quite like this: given some additional memory, the gathers can be "double-buffered" by fetching subsequent layers while performing computation. Some implementations, e.g., PyTorch's, are able to do this.
7. Minor: The paper uses "words per second" for throughput; is this really tokens per second?

**Questions:**

Please see the comments/questions above under "weaknesses".

Please focus in particular on whether there are actionable insights beyond the limits to strong scaling and the guidelines in Narayanan et al.; I would be willing to raise my score if a convincing case is made for this.

---

> ### Author Response · Authors · 2024-11-27
>
> We greatly appreciate the reviewer’s insights, provided references, and detailed feedback! We acknowledge that the emergence and dominance of communication overhead is known behavior in parallel computing. Nonetheless, our work provides effective guidance for the machine learning community of researchers and practitioners by providing empirical results demonstrating these scaling patterns.
>
> We demonstrate common regimes in which communication overhead dominates: e.g. the exact regimes in which over-decomposition of a 7B and 70B models on commodity hardware results in communication exceeding compute. Additionally, we highlight that the preference to model parallelism over data parallelism is a distinct conclusion to prior work comparing parallelization strategies.

---

> > ### Author Response · Authors · 2024-11-27
> >
> > Below, we address the reviewer’s comments and concerns.
> > 1. **Weak and Strong Scaling.** In contrast to Narayanan et. al 2021, which experimentally shows that Megatron-LM performs well under weak scaling of model size (i.e. increasing model size and device world size jointly), our work provides empirical experiments showing diminishing returns in distributed training from both weak and strong scaling of the data batch sizes with number of devices.
> >  * Weak Scaling (Sec 4.1 and Appendix E): We study weak scaling of increasing global batch size with increasing numbers of processors. Weak scaling in this manner is manner is representative of training regimes where a variable number of processors may be used at cost of increased per iteration latency with techniques such as gradient accumulation.
> >
> > * Strong Scaling (Sec 4.3 and Appendix C):  Although diminishing returns in strong scaling are expected, we observe that over decomposition emerges when degree of parallelism yields an effective local batch size of less than a single example (Fig 7).
> >
> > 2. **Comparisons with Megatron Studies.** In Narayanan et al 2021 and and Hagemann et. al 2024, tensor and pipeline parallelism are the primary methods for reducing memory pressure; neither use sharding via FSDP or Zero [1,2]. Both works concludes that the frequent all-to-all communication of tensor parallelism is suboptimal to increased data parallelism due to blocking communication in the critical path and increased frequency of communication. \
> > \
> > Additionally, Narayanan et al 2021 concludes that Megatron 3D parallelism outperforms Zero-3 without model parallelism, analagous to FSDP, and speculates that model parallelism may be able to improve utilization of Zero based methods but does not provide experimental results. \
> > \
> > By contrast, we consider FSDP with 3D parallelism as approaches to alleviate memory pressure. In this setting, model parallelism is not needed as parameter and optimizer state sharding sufficiently reduces memory pressure – yet we show introduction of model parallelism is still beneficial, contradicting prior work. \
> > \
> > In contrast to studies with Megatron, we would like to highlight that in our work we conduct additional analysis utilizing multiple generations of GPU hardware, which empirically show the worsening trends in communication overhead due to disproportionate increases in compute speed relative to internode fabric; and additional reporting of GPU power utilization which exhibits negative scaling behaviors.
> >
> > 3. **Concerns over Evaluated Configurations.** Hybrid-Sharded Data Parallelism has been introduced as a mitigation for the communication costs of FSDP and Zero sharding and has been used in the training of some public LLMs (e.g. DataBricks MoE). However, vanilla fully sharded data parallelism with 3D parallelisms is still a representative approach to training many models and is currently used in the training of multiple large language models at large-scale (e.g. Meta Llama 3.1, ByteDance MegaScale, IBM Granite 3.0, AI2 Olmo) [3,4,5,6].
> >
> >
> > **Other Concerns**
> > * **Algorithms for AllGather and ReduceScatter.**  We appreciate the provided references, and acknowledge the oversight in our writing and will clarify in revision. The statement was in reference to the implementations of AllGather and ReduceScatter in NCCL, as used in Megatron and our own framework.  To the best of our knowledge and based on public source code, ReduceScatter and AllGather only supports Ring algorithms (ref: https://github.com/NVIDIA/nccl/blob/master/src/device)
> >
> > * **Buffering of FSDP AllGathers.** In our experiments, our implementation of FSDP the mentioned AllGather prefetch operations during prior layer computation. However, due to the large cost of the AllGather collective over large numbers of devices, the operation still leads to exposed communication at larger scale. We will clarify our implementation in revision.
> >
> > * Yes, the words per second measurement corresponds to tokens per second.
> >
> > **References**
> > 1. Narayanan et al., "Efficient Large-Scale Language Model Training on GPU Clusters Using Megatron-LM", Supercomputing 2021
> > 2. Hagemann, Johannes, et al. "Efficient Parallelization Layouts for Large-Scale Distributed Model Training." Workshop on Advancing Neural Network Training: Computational Efficiency, Scalability, and Resource Optimization (WANT@ NeurIPS 2023). 2023.
> > 3. Dubey, Abhimanyu, et al. "The llama 3 herd of models." arXiv preprint arXiv:2407.21783 (2024).
> > 4. Jiang, Ziheng, et al. "{MegaScale}: Scaling large language model training to more than 10,000 {GPUs}." 21st USENIX Symposium on Networked Systems Design and Implementation (NSDI 24). 2024.
> > 5. Hess, Peter. “Training AI Models Faster than Ever.” IBM Research, IBM, 18 Sept. 2024, research.ibm.com/blog/pytorch-2024-training-models. Accessed 27 Nov. 2024.
> > 6. Groeneveld, Dirk, et al. "Olmo: Accelerating the science of language models." arXiv preprint arXiv:2402.00838 (2024).

---

### Note · Authors · 2024-12-05

**Comment:**

We would like to thank the reviewers for their comments and feedback, and will be withdrawing the submission.

**Withdrawal Confirmation:**

I have read and agree with the venue's withdrawal policy on behalf of myself and my co-authors.